# A hybrid CNN-SVM model for enhanced autism diagnosis

Linjie Qiu⬡*, Jian Zhai⬡

School of Mathematical Sciences, Zhejiang University, Hangzhou, Zhejiang, China

⬡ These authors contributed equally to this work.
* 11935034@zju.edu.cn

**Data Availability Statement:** The datasets we used are common public datasets from URL: http://fcon_1000.projects.nitrc.org/indi/abide/. The codes associated with this study available at: https://github.com/jackykyu/A-hybrid-CNN-SVM-model-for-enhanced-autism-diagnosis.git.

## Abstract

Autism is a representative disorder of pervasive developmental disorder. It exerts influence upon an individual's behavior and performance, potentially co-occurring with other mental illnesses. Consequently, an effective diagnostic approach proves to be invaluable in both therapeutic interventions and the timely provision of medical support. Currently, most scholars' research primarily relies on neuroimaging techniques for auxiliary diagnosis and does not take into account the distinctive features of autism's social impediments. In order to address this deficiency, this paper introduces a novel convolutional neural network-support vector machine model that integrates resting state functional magnetic resonance imaging data with the social responsiveness scale metrics for the diagnostic assessment of autism. We selected 821 subjects containing the social responsiveness scale measure from the publicly available Autism Brain Imaging Data Exchange dataset, including 379 subjects with autism spectrum disorder and 442 typical controls. After preprocessing of fMRI data, we compute the static and dynamic functional connectivity for each subject. Subsequently, convolutional neural networks and attention mechanisms are utilized to extracts their respective features. The extracted features, combined with the social responsiveness scale features, are then employed as novel inputs for the support vector machine to categorize autistic patients and typical controls. The proposed model identifies salient features within the static and dynamic functional connectivity, offering a possible biological foundation for clinical diagnosis. By incorporating the behavioral assessments, the model achieves a remarkable classification accuracy of 94.30%, providing a more reliable support for auxiliary diagnosis.

## Introduction

Autism spectrum disorder (ASD) is a neurodevelopmental disease marked by language impairment, social impairment, and stereotyped behavior. According to statistical data, approximately one in every 100 children is diagnosed with autism [1]. Despite its characteristics often being identifiable in early childhood, definitive diagnosis typically requires an extended period of time. Given that individuals with ASD generally need specialized medical care to mitigate the risk of co-occurring mental health conditions, timely and accurate diagnosis is imperative.

**Funding:** This work was supported by the National Natural Science Foundation of China under Grant numbers 11671354. The funders had no role in study design, data collection and analysis, decision to publish, or preparation of the manuscript.

Nowadays, machine learning is widely employed in auxiliary diagnosis across various medical conditions, such as brain tumors [2, 3], autism [4, 5], depression [6] and more. They learn features from data and help doctors screen and diagnose diseases. These methods include traditional machine learning techniques, such as support vector machine (SVM) [7], random forest (RF), and deep learning models, such as convolutional neural network (CNN) [8] and vision transformer (ViT) [9]. However, compared to doctors who make comprehensive judgments based on all aspects of information, most models tend to rely on a certain aspect of subjects, such as patients' medical imaging data. When dealing with conditions characterized by lesions, such as brain tumors, a reliance on brain imaging data proves effective [10]. However, for mental disorders like autism, a singular data source may be insufficient. Therefore, a more rational approach involves the integration of multiple facets through an ensemble learning model.

In the context of autism, numerous studies have proposed various models from different perspectives, offering diverse strategies for diagnosis. There are classification methods based on extracting features from facial image data [11], classification based on extracted features from audio and video data of subjects' behavior [12], classification based on subjects' eye-tracking data [13], and models based on functional magnetic resonance imaging (fMRI) data [5], etc. Among these, fMRI technology has gained widespread usage in research due to its high resolution and non-invasive nature, both in task-based and resting-state scenarios [14, 15]. Based on fMRI technology, there are models that directly utilize MRI data for deep learning construction [16] and others that employ various preprocessing tools to extract features for subsequent analysis. Functional connectivity (FC) is a commonly used analytical tool. The intricate physiological activities within the human body orchestrated through the coordinated interplay among various brain regions, and the relationship between brain regions evolve over time. Leveraging these connections holds promise for diagnosis of autism and finding areas that cause brain disorders. Resting-state functional magnetic resonance imaging (rs-fMRI), with its advantages of not requiring additional complex tasks, has been widely used in previous study. Constructing functional connectivity based rs-fMRI data serves as an efficient method for characterizing relationships among brain areas. Early scholarly articles often focused solely on static functional connectivity, which presupposes that brain interrelations remain constant throughout the whole scanning process. However, recent literature reveals that neural connections fluctuate during the scan, thus giving rise to the broader application of dynamic functional connectivity in brain activity research [17–19]. To furnish a more comprehensive evaluation of the brain activities, this paper employs both static and dynamic functional connectivity to analyze differences between the subjects with ASD and typical controls (TCs), trying to identify areas causing ASD. Nevertheless, only the analysis of brain activity may prove.

As mentioned earlier, relying solely on the analysis of brain activity connections may not suffice for the effective diagnosis of psychiatric disorders. Furthermore, since autism mainly manifests impairments in social behavior, FC may not adequately reflect this aspect. To address this limitation, more information is required. Current psychiatric screening or diagnostic procedures predominantly rely on behavioral observations rooted in symptomatology. For example, the Social Responsiveness Scale–Second Edition (SRS-2) serves as an early screening tool for ASD [20], quantifying social functionality in five aspects: awareness, cognition, communication, motivation, and mannerisms. Substantial evidence supports its effectiveness and sensitivity in identifying autism symptoms in school-age children. However, the questionnaire mainly focuses on social communication, and only a few items involve stereotyped behavior. Researchers are still gathering more data on the stability and effectiveness of the SRS-2 [21–23]. While other diagnostic tools such as the Autism Diagnostic Observation Schedule, the Childhood Autism Rating Scale, and the Autism Diagnostic Interview-Revised

are commonly employed in clinical practice, these methods are more time-consuming. However, these methods are inevitably affected by clinical training, tools, and cultural background, thereby interfering with clinicians' subjective observations [24, 25]. To overcome this limitation, it is not enough to only use the scale results to assist diagnosis. Consequently, we have integrated analyses of functional connectivity to better assist diagnosis.

To harness the full potential of early screening questionnaires and brain functional connectivity for autism diagnosis, this paper introduces a hybrid CNN-SVM model. This model employs a CNN architecture to extract deep features from both static and dynamic functional connectivity. In the learning process, the feature extraction based on the frequency band and the convolution kernel based on the functional connectivity matrix are used, and the attention mechanism is introduced to weight the learned features. Finally, the learned features combined with the features from the SRS are sent to SVM for classification. After studying 379 subjects with ASD and 442 typical controls, our model achieves good classification accuracy and provides the most discriminating brain regions and scanning bands. The contributions of this work are summarized as:

1. A novel ensemble learning model is proposed, which integrates information from FC in fMRI data with the SRS scores. SRS scores reflect the degree of social impairment in subjects.

2. Leveraging fMRI data, we construct both static and dynamic FC. Based on the CNN structure of extracted features, the brain areas and frequency bands that play a greater role in classification are found. And the attention mechanism is introduced to enhance the extracted features from FC.

3. To assess performance, we conduct comparative evaluations with various machine learning models, RF, logistic regression (LR), multilayer perceptron (MLP) and SVM. We also show the model's performance on data across genders, age groups and sites. Besides, we compare our model with some previous works.

The structure of this paper is as follows. Section 2 introduces data preprocessing and network architecture. Section 3 presents our results. Section 4 discusses the limitations of our study and the future work. Section 5 provides the conclusions.

## Materials and methods

The whole process of this experiment is illustrated in Fig 1. More details of Data preprocessing and CNN for feature extraction are shown in the following sections.

To enhance the comprehension of Fig 1, we shall describe the flowchart in algorithmic form as follows:

Input:

- Training dataset $D = \{(x_1, y_1), (x_2, y_2), \ldots, (x_n, y_n)\}$, where $x_i$ denotes features, and $y_i$ represents the corresponding labels.

- The number of folds for cross-validation, denoted as $k = 10$.

Output: A well-trained model.
Algorithm Description:

1. Preprocess the fMRI data $x_i$ from the training dataset $D$, following the steps outlined in the subsection 0.1. This yields a new dataset $D_1 = \{(a_1, t_1, y_1), (a_2, t_2, y_2), \ldots, (a_n, t_n, y_n)\}$, where $a_i$ represents SRS features, $t_i$ corresponds to the ROI time series, and $y_i$ denotes labels.

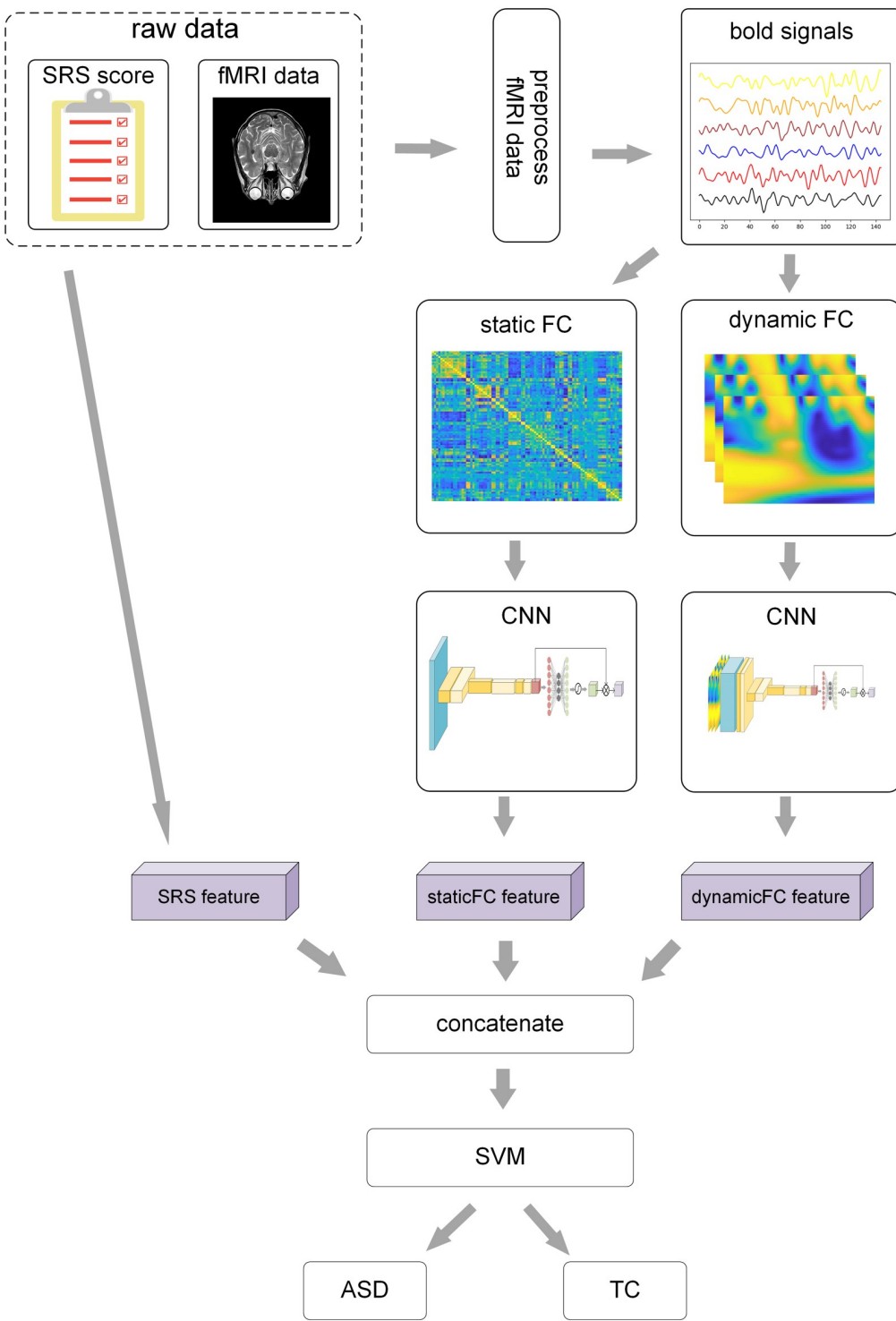

**Fig 1. The entire process of this experiment.** Details of the CNN for extracting features from static and dynamic FC parts are illustrated in Figs 3 and 4 separately. The left figure in the raw data part is republished from [26] under a CC BY license, with permission from Pixabay, original copyright 2017. The right figure in the raw data part is republished from [27] under a CC BY license, with permission from Pixabay, original copyright 2015.

2. Based on $t_i$, calculate the required static FC $b_i$ and dynamic FC $c_i$, with detailed computational formulas provided in subsections 0.2 and 0.3. This results in a new dataset $D_2 = \{(a_1, b_1, c_1, y_1), (a_2, b_2, c_2, y_2), \ldots, (a_n, b_n, c_n, y_n)\}$.

3. Partition the training dataset $D_2$ into $k$ non-overlapping subsets, typically employing a K-fold cross-validation approach. For each fold $i$, designate it as the validation set while using the remaining $k - 1$ folds as the training set.

4. Feed the static FC $b_i$ and dynamic FC $c_i$ of the training set into a CNN for deep feature extraction. Utilize the backpropagation algorithm and Adam optimizer to update model parameters. Employ the cross-entropy loss function, monitor its convergence during training, and set the maximum number of epochs to 50. The details of neural network architecture can be found in subsection 0.4. Assuming the extracted deep features are denoted as $d_i$ and $e_i$, combine them with $a_i$ as $(a_i, d_i, e_i)$ and feed them into a linear kernel SVM for training.

5. Utilize the well-trained model to make predictions on the validation set and compute performance metrics.

6. Repeat steps 4 and 5 iteratively, using each fold as the validation set, to obtain a set of performance metrics.

7. Calculate the average performance metrics from the $k$ folds to assess the model's performance. Concurrently, save the model parameters from the training process with the best performance metrics as the final model for deployment.

## 0.1 Data preprocessing

Firstly, we select resting-state fMRI data that includes SRS metrics from the publicly available Autism Brain Imaging Data Exchange (ABIDE) dataset [28]. The distribution and age of the subjects selected are shown in the Table 1 and Fig 2. Upon acquiring the data from the ABIDE dataset, we perform the following preprocessing steps:

**Table 1. Distribution of the data from rs-fMRI ABIDE database used in this study.**

| Site | ASD | TC | Total | AgeRange |
|------|-----|-----|-------|----------|
| LEUVEN | 29 | 35 | 64 | 12-32 |
| BNI | 28 | 29 | 57 | 18-64 |
| ETH | 11 | 22 | 33 | 13-31 |
| GU | 51 | 55 | 106 | 8-14 |
| IU | 15 | 11 | 26 | 17-54 |
| KKI | 52 | 148 | 200 | 8-13 |
| NYU | 75 | 28 | 103 | 5-34 |
| OHSU | 33 | 46 | 79 | 8-15 |
| SDSU | 32 | 25 | 57 | 7-18 |
| TCD | 21 | 21 | 42 | 10-20 |
| UCD | 18 | 14 | 32 | 12-18 |
| USM | 14 | 8 | 22 | 9-39 |
| total | 379 | 442 | 821 | |

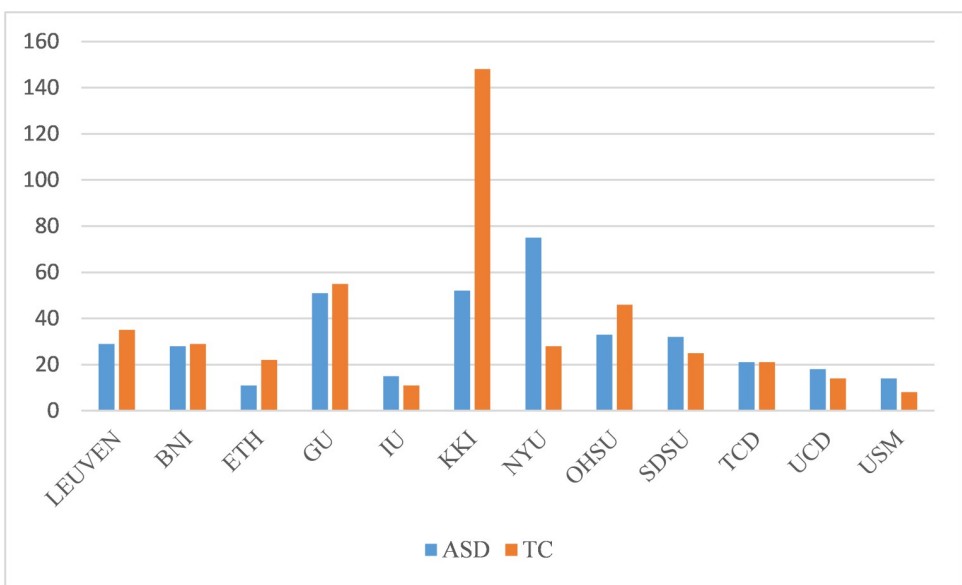

**Fig 2. The distribution of subjects in the dataset.**

1. Extract the evaluation scores of the five aspects of the SRS, convert them into a one-dimensional vector and subsequently send it to the final SVM for classification, as depicted in the SRS feature in Fig 1.

2. Remove the first ten time points of each subject to mitigate errors caused by the instability of the gradient magnetic field at the beginning of the scan; correct slice scan time to ensure that the resampled data can be seen as scanned at the same time point; carry out head motion correction to mitigate the impact of head movements to some extent; and perform nuisance regression to eliminate the influence of extraneous factors.

3. Register the structural image data and then map it to the standard space-MNI space, which aligns the images of each subject to ensure that the anatomical structures of different subjects correspond to the same voxels, and perform spatial smoothing to enhance registration effects.

4. Employ band–pass filtering (0.01–0.08 Hz). Most nuclear magnetic resonance signals are low–frequency signals. Given that magnetic resonance signals are primarily low–frequency signals, this frequency range is chosen to minimize the impact of physiological noise from cardiac ($\sim$ 0.15) and respiratory ($\sim$0.3 Hz) activities [29].

5. Utilize the Automated Anatomical Labeling (AAL) atlas, which partitions the brain into 116 regions. These regions serve as our regions of interest (ROIs), and the corresponding time series are extracted.

This study uses SPM12 and DPARSF [30] to preprocess fMRI data as mentioned above, and the extracted time series is used for subsequent processing.

## 0.2 Static functional connectivity

The construction of a static functional connectivity matrix is predicated on the Pearson correlation coefficients between various regions of interest in the brain, as shown in Eq 1.

$$R(X, Y) = \frac{\sum (x_i - \bar{x}_i)(y_i - \bar{y}_i)}{\sqrt{\sum (x_i - \bar{x}_i)^2} \sqrt{\sum (y_i - \bar{y}_i)^2}},$$

(1)

where X and Y are two distinct brain regions, while $x_i$ and $y_i$ denote the corresponding time series. $\bar{x}_i$ and $\bar{y}_i$ are mean values of $x_i$ and $y_i$ separately. The underlying assumption of static functional connectivity is that the interconnections between different brain areas remain invariant throughout the entire scanning procedure, and the Pearson correlation coefficient characterizes this relationship [18]. The resulting static functional connectivity matrix has dimensions of 116 × 116, as illustrated in the input part of Fig 3.

## 0.3 Dynamic functional connectivity

Throughout the scanning process, the functional connectivity between different brain regions exhibits temporal fluctuations, indicating that a representation solely based on static functional connectivity would be overly simplistic. This has led to the introduction of dynamic functional connectivity, which also serves to enrich the data set. Sliding-window analysis is a method often considered for dynamic functional connectivity. It describes changes in the brain's connection patterns by using a fixed window size and moving it at a specific stride. However, there is no consensus on the optimal size for the fixed temporal window. Additionally, variations have been observed in connectivity matrices at different frequencies [31, 32]. These issues cannot be solved by sliding-window method. We therefore perform a time-frequency analysis of the signal. We choose to use wavelet analysis to characterize dynamic functional connectivity, which effectively extracts information from signals through operations such as scaling and translation, thus enabling a multi-scale, fine-grained examination.

Initially, we employ continuous wavelet transforms to process the time series for each brain area, as delineated by the Eq 2.

$$W(s, \tau) = \frac{1}{\sqrt{s}} \int_{-\infty}^{\infty} x(t) \phi^* \left( \frac{t - \tau}{s} \right) dt,$$

(2)

where $s$ is the wavelet scale, $\tau$ denotes the translation value, $x(t)$ is the signal, $\phi$ represents a mother wavelet and $^*$ denotes the complex conjugate [33]. Scale $s$ controls the expansion and contraction of the wavelet function, which is inversely proportional to frequency. The translation $\tau$ controls the translation of the wavelet function, which corresponds to time points. When we traverse the required scales and translations, we can obtain the needed spectrogram from the unstable signal for subsequent analysis. We employ the Morlet wavelet as the mother wavelet, which offers an optimal ratio between frequency bands and wavelet scales, facilitating the understanding of data within specific frequency ranges [34]. In our analysis, we partition the frequency range of 0.01-0.08 Hz into 40 segments, and the translation $\tau$ corresponds to the sampled time points.

After obtaining the time-frequency spectra for each region of interest, we proceed to compute their dynamic functional connectivity. Here, in order to describe the time-frequency spatial relationship between two signals, we initially introduce the concept of cross-wavelet power [33], which identifies regions of the common high power between the two signals, as shown in

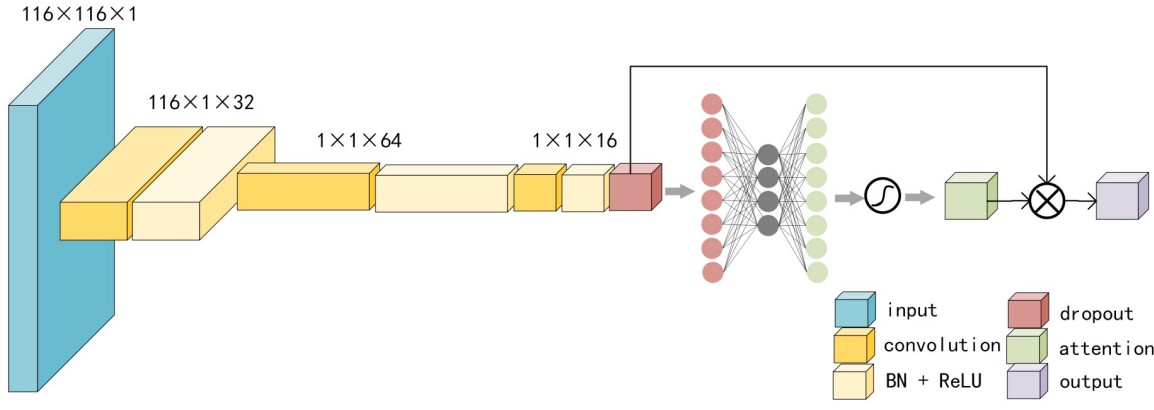

**Fig 3. The entire process of processing static functional connectivity.**

Eq 3.

$$CW_{xy}(s, \tau) = W_x(s, \tau) W_y^*(s, \tau), \tag{3}$$

where $W_x$ and $W_y$ are the continuous wavelet of $x$ and $y$ separately. To further describe the coherence of these two cross wavelets in time-frequency space as the functional connectivity between two brain areas, we introduce the concept of wavelet coherence [33], calculated as Eq 4.

$$R_{xy} = \frac{|S(CW_{xy}(s, \tau))|^2}{S(|W_x(s, \tau)|^2) \cdot S(|W_y(s, \tau)|^2)}, \tag{4}$$

where $S(\cdot)$ denotes a smoothing operator related to scale and time, for which we employ the most commonly used moving average technique.

Finally, after processing different times and scales, we convert the signals of any two regions of interest into a matrix $WC \in R^{m \times n}$, here $m$ corresponds to the number of chosen scales, specifically $m = 40$, and $n$ corresponds to the number of time points. In this way, for the same individual, we can get 6612 completely different matrices. Obviously, the amount of data is too large. At the same time, the data obtained by different institutions have different time points $n$, and so they cannot be easily passed to the same neural network for learning. To this end, we consider employing Principal Component Analysis (PCA) on the matrices [35]. In order to facilitate the unification of data from different institutions, we opt to perform dimensionality reduction on the temporal dimension, as detailed in the following procedures:

1. Calculate wavelet coherence for two regions of interest to obtain a matrix $WC = [wc_1, \ldots, wc_n]$

2. For the matrix $WC$, remove the average value to get $\hat{W}C = [w_1, \ldots, w_n]$, where $w_i = wc_i - \sum_{j=1}^{n} wc_i / n, i = 1, 2, \ldots, n.$

3. Construct matrix $\frac{1}{n} \hat{W}C \cdot \hat{W}C^T$. Use the eigenvalue decomposition method to calculate its eigenvalues and the corresponding unit eigenvectors. Suppose there are $l$ eigenvalues and unit eigenvectors, and eigenvalues $\lambda_1, \lambda_2, \ldots, \lambda_l$ are arranged from large to small. Then $\frac{1}{n} \hat{W}C \hat{W}C^T X_i = \lambda_i X_i, i = 1, 2, \ldots, l.$

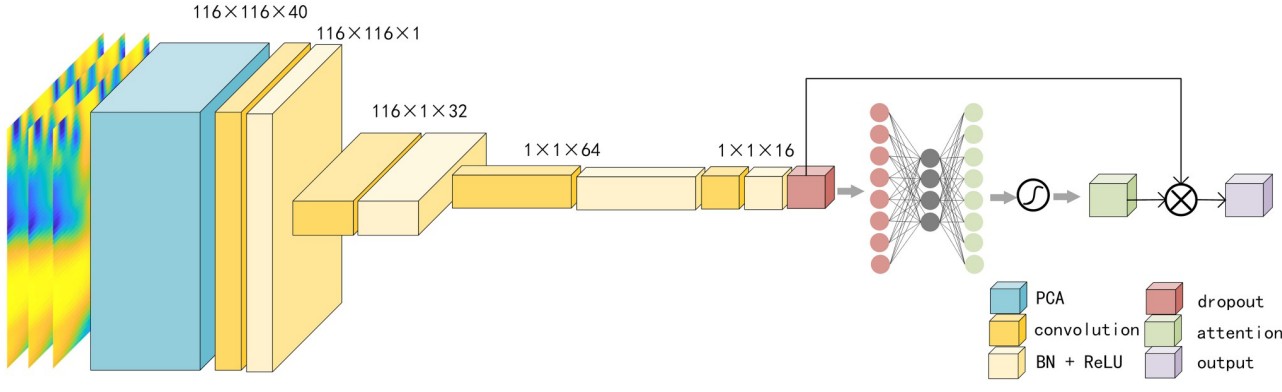

**Fig 4. The entire process of processing dynamic functional connectivity.**

4. In order to retain 99% of the information, it is necessary to find the smallest k that satisfies $\frac{\sum_{i=1}^{k} \lambda_i}{\sum_{i=1}^{l} \lambda_i} \geq 0.99$. These k unit eigenvectors can form an eigenvector matrix P.

5. Finally, the matrix is mapped to the space composed of the corresponding feature vectors, that is, $XP \in R^{m \times k}$.

After performing principal component analysis on all matrices $WC \in R^{m \times n}$, we find that $k = 1$ can satisfy the condition in step 4. Thus we finally obtain a matrix of $116 \times 116 \times 40$ for each subject as the dynamic functional connectivity.

## 0.4 CNN for feature extraction

As illustrated in Fig 1, the input to our model is based on data from SRS tables, static FC and dynamic FC. Here, we employ CNNs to extract features from static FC and dynamic FC. For ordinary pictures, feature extraction is proficiently achieved through the utilization of $3 \times 3$ or $5 \times 5$ convolution kernels [36, 37]. However, when it comes to the FC matrix for a subject, each row or column of the matrix corresponding to the correlation between two brain areas. The aforementioned convolution kernel cannot summarize such information, and there is no reasonable explanation for the local features. Therefore, we believe that a more reasonable convolution kernel should be $1 \times n$ or $n \times 1$, where $n$ corresponds to the number of regions of interest, that is, 116. The convolution kernel proposed here is more capable of extracting brain areas corresponding to functional connectivity that make greater contributions to classification.

For static FC, each subject corresponds to a $116 \times 116 \times 1$ tensor, as shown in Fig 3. We send it into the CNN. The first layer comprises 32 filters, and the corresponding convolution kernel is $1 \times 116$. After passing through the batch normalization (BN) layer and rectified linear unit (ReLu) layer, it is sent to the second convolution layer. The second layer comprises 64 filters, and the corresponding convolution kernel is $116 \times 1$. After passing through the BN layer and ReLu layer, it is sent to the third convolution layer. The third layer comprises 16 filters, and the corresponding convolution kernel is $1 \times 1$. Finally, it goes through the BN layer and ReLu layer. The third layer of convolutional layer is to prevent the extracted feature dimension from being too large and affecting subsequent classification. In addition to this step, we also use a dropout layer to prevent overfitting, where the dropout rate is set to 50%. All activation functions employed are of the LeakyReLU type ($\alpha = 0.01$).

The most discriminative deep features need to be distinguished through the attention mechanism [38], which follows the dropout layer. This attention network is bifurcated into two fully connected layers: the first boasting 8 neurons, followed by a ReLU layer, and the second containing 16 neurons, which is the same number of the input neurons. Then, it passes through the Sigmoid function to obtain the normalized weight of each input feature. These normalized weights are utilized for weighted summation to extract the requisite static functional connectivity features. To facilitate backpropagation for parameter tuning, an additional layer is added, the size of which corresponds to the number of classification categories, which is 2. The loss between the actual labels and the predicted outcomes is minimized via the adaptive moment estimation optimizer, with a learning rate set at 0.0001. The employed loss function is cross-entropy. However, the focus is not on the network's predictive capacity. We just use it to train the network parameters and so the final hidden layer's parameters serve as the learned static functional connectivity features, as shown in the staticFC feature in Fig 1.

For dynamic FC, we can know that each object corresponds to a matrix of $116 \times 116 \times 40$ after PCA processing, as shown in Fig 4. The first step is channel compression. As mentioned before, each $116 \times 116$ layer corresponds to a distinct frequency. The compression of channels can be interpreted as the neural network looking for the most discriminative frequency bands for classification, which may have certain reference value for the analysis of fMRI data. Hence, in comparison to the extraction of features from static FC, the processing of dynamic FC necessitates an additional step of channel compression. The first layer comprises 1 filter with a corresponding $1 \times 1$ convolutional kernel. Then it is sent to the BN layer and the ReLu layer. The subsequent operations is similar to the operations that performed on the static FC. Finally the dynamicFC features required in the Fig 1 are obtained.

Subsequently, SRS features are concatenated with those gleaned from both static FC and dynamic FC. This composite feature set is then submitted to a SVM with a linear kernel for further classification. This experiment uses a 10-fold cross-validation of the data. The reason why this study uses SVMs instead of the multi-layer perceptron (MLP) for classification is because in experiments we find that the MLP model has poor generalization capability and performs poorly on the test set. In the support vector machine part, we use the more common linear kernel for classification, because the extracted feature dimension is small enough compared to the data set. In order to illustrate the advantages of SVM, we also conduct experiments using MLP and RF for classification. The architecture of the MLP consists of two hidden layers, featuring 64 and 16 neurons respectively. The number of neurons in the input layer is the dimension after concatenating three features, and the output layer is 2 neurons. As for RF, 100 trees are selected for classification. And for LR, we choose a threshold of 0.5 to distinguish subjects with ASD and TCs. These experiments all adopt the 10-fold cross-validation, and the results are averaged after several experiments.

## Results

### 0.5 Classification comparison

According to the results of cross-validation, sensitivity (SEN), specificity (SPE), accuracy (ACC), false positive rate (FPR), false negative rate (FNR) and F1 Score (F1) are used as evaluation indicators for classification results. Here, true positives (TPs) are considered to be correctly diagnosed ASD patients, true negatives (TNs) are considered to be truly diagnosed TCs, false positives (FPs) and false negatives (FNs) are respectively incorrectly diagnosed ASD

patients and TCs. SEN, SPE and ACC are calculated as follows,

$$ACC = \frac{TP + TN}{TP + FP + TN + FN} \tag{5}$$

$$SEN = \frac{TP}{FN + TP} \tag{6}$$

$$SPE = \frac{TN}{FP + TN} \tag{7}$$

$$FPR = \frac{FP}{FP + TN} \tag{8}$$

$$FNR = \frac{FN}{FN + TP} \tag{9}$$

$$F1 = 2\frac{Precision * SEN}{Precision + SEN}, \tag{10}$$

where $Precison = TP/(TP + FP)$. Precision focuses on the model's accuracy in predicting positive examples. In order to balance the precision and SEN, the F1 score are used to comprehensively evaluate model performance. The ACC shows the model's ability to predict correctly, while SEN and SPE tell the model's ability to identify subjects and negative examples respectively. The FPR refers to the rate at which the model incorrectly classifies actual negative examples as positive examples. The FNR refers to the rate at which the model incorrectly classifies actual positive examples as negative examples. After ten-fold cross-validation, our framework achieves 94.30% ACC, 92.87% SEN, 95.47% SPE, 7.12%FPR, 4.52% FNR and 94.73% F1. Fig 5 illustrates the performance of different classifiers.

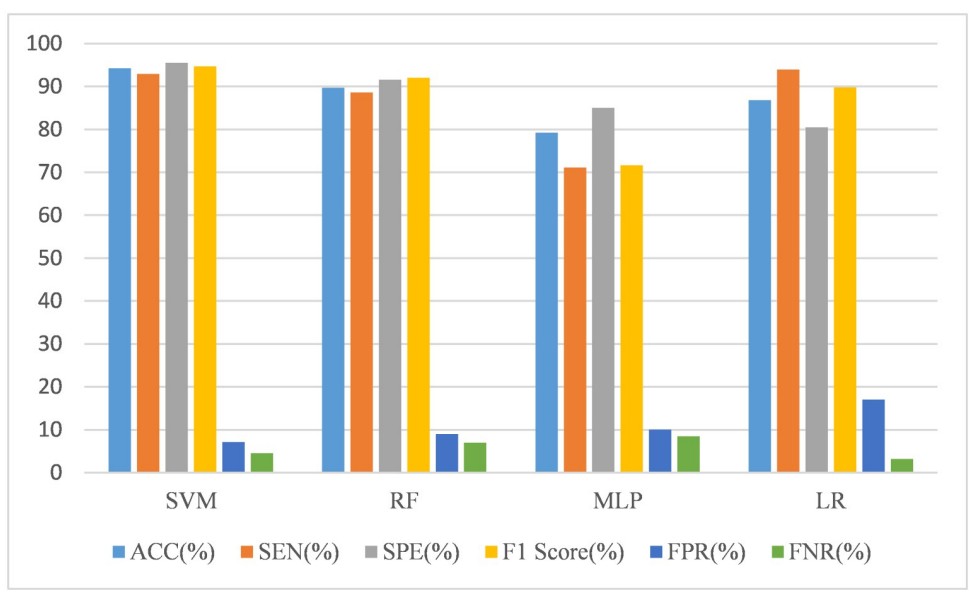

**Fig 5. The performance of different classifiers.**

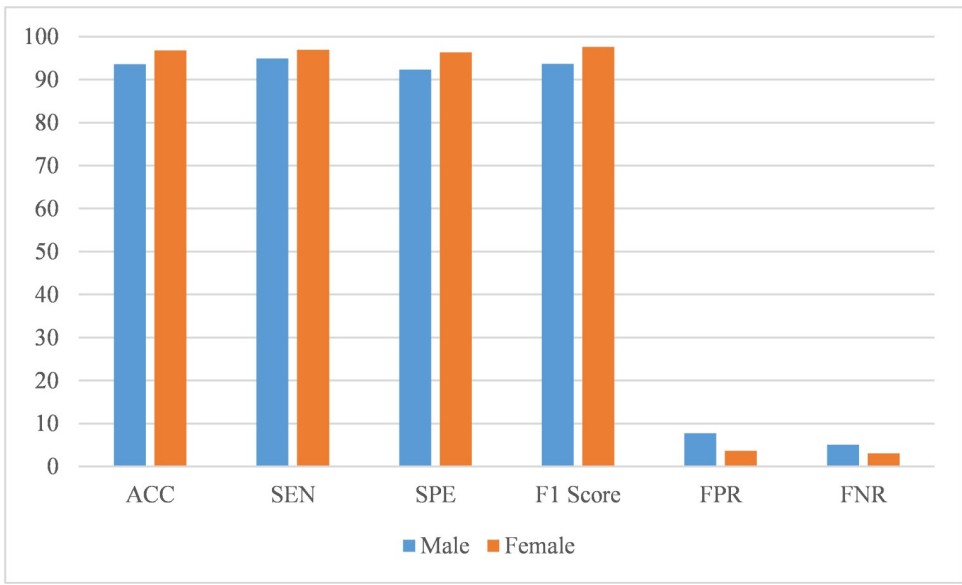

**Fig 6. The performance of the CNN-SVM model on different gender.**

Since the classification performance of the model can be affected by gender and age, we also analyze these two variables according to the results, as shown in Figs 6 and 7.

From Fig 6, we can see that the overall performance of women is better than men when the model makes predictions. The dataset used in this article involves 175 women and 646 men. This is because the prevalence rate in men is four times that of women, which may lead to an imbalance in the collected data. Thus, further confirmation of whether the model is better at

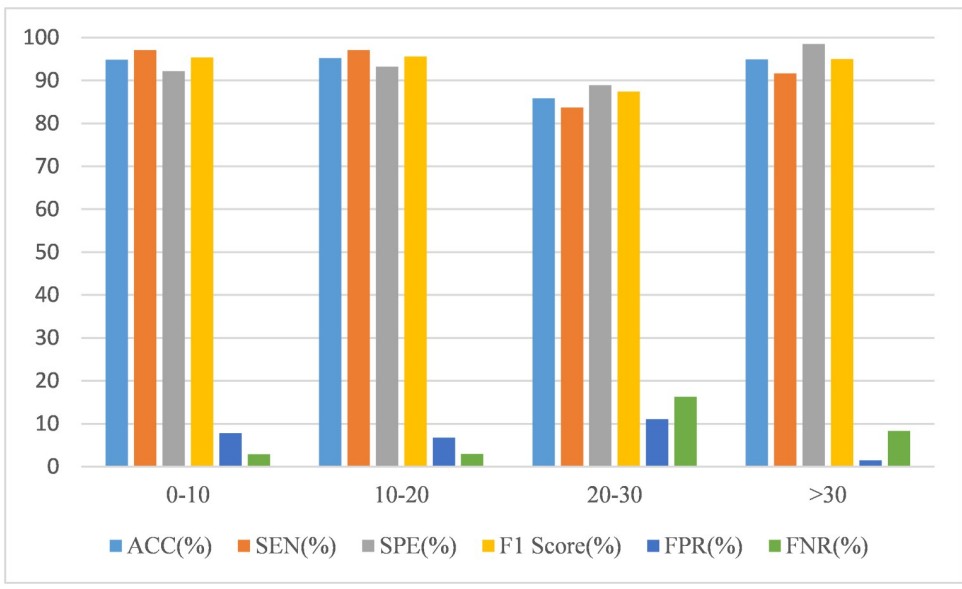

**Fig 7. The performance of the CNN-SVM model on different age range.**

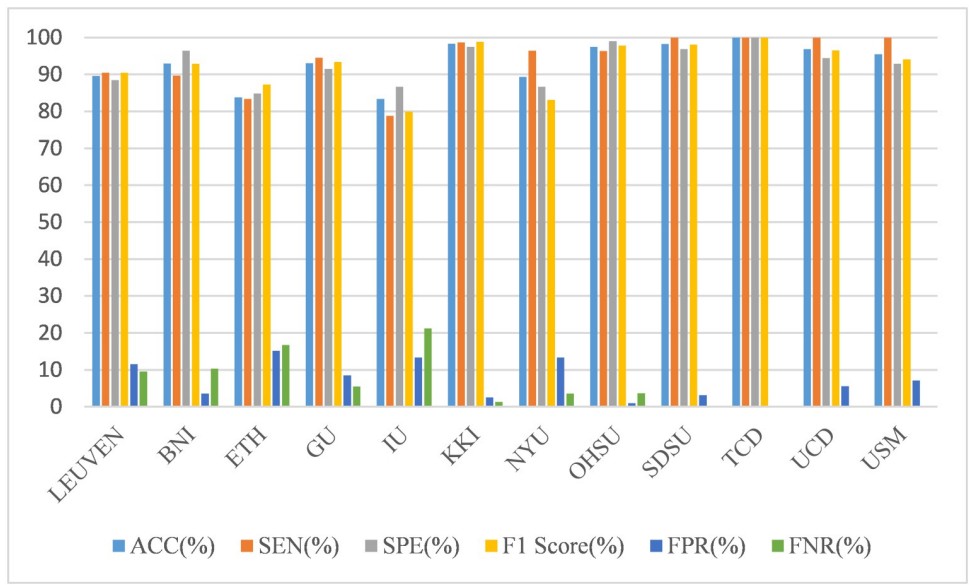

**Fig 8. The performance of the CNN-SVM model on data from different sites.**

predicting female subjects than men requires collecting more data. Overall, the performace of our model does not differ significantly across gender.

From Fig 7, we can see that the model's prediction results are pretty good in multiple age groups, and its performance is only slightly worse for subjects aged 20-30. This means that our model is less affected by age.

At the same time, we also compare the prediction results of the model on data from different sites. As shown in Fig 8, we find that the model perform relatively well at most sites, but the performance is not very satisfactory at only two sites, ETH and IU. It shows that our model has a certain degree of robustness when processing data from different sites.

To measure the performance of our model, we compare it with some models proposed in previous papers, as shown in Table 2.

**Table 2. Performance comparison of multiple papers based on ABIDE dataset.** DNN: Deep neural network; DBN: Deep Belief Network; CNNG:convolutional neural network and gate recurrent unit; BNC-DGHL:a brain network classification method based on deep graph hashing learning; C-GAN:Conditional Generative Adversarial Network; A-GCL: an adversarial self-supervised graph neural network based on graph contrastive learning; AWSO: the Adam war strategy optimization.

| Paper | Feature | Model | *ACC* | *SEN* | *SPE* |
|---|---|---|---|---|---|
| [39] | static FC | DNN | 93.17% | 91.21% | 94.93% |
| [40] | static FC | DBN | 76.40% | 76.20% | 73.10% |
| [41] | fMRI images | CNN | 71.81% | 81.25% | 68.75% |
| [16] | fMRI images | CNNG | 72.46% | 71.35% | 79.25% |
| [42] | static FC | BNC-DGHL | 77.1% | 77.8% | 75.8% |
| [43] | static FC | MLP | 74.82% | 67.33% | 80.75% |
| [44] | texture feature, dynamic FC | C-GAN | 74.00% | 79.00% | 74.00% |
| [45] | ALFF, static FC | A-GCL | 80.65% | - | 82.28% |
| [46] | fMRI images | AWSO-DBN | 92.40% | 93.00% | 93.50% |
| this paper | SRS score, static FC, dynamic FC | CNN-SVM | 94.30% | 92.87% | 95.47% |

**Table 3. The discriminating brain areas of static FC and dynamic FC.**

| Order | Static FC | Dynamic FC |
|---|---|---|
| 1 | SupraMarginal_L | Cerebellum_Crus2_R |
| 2 | Cingulum_Post_L | Thalamus_R |
| 3 | Cerebelum_Crus1_R | Cerebellum_3_R |
| 4 | Heschl_R | Heschl_R |
| 5 | Occipital_Inf_R | Temporal_Mid_L |
| 6 | Cerebelum_7b_L | Frontal_Inf_Oper_R |
| 7 | Cerebelum_9_R | Precentral_R |
| 8 | Cerebelum_3_R | Vermis_6 |
| 9 | Frontal_Sup_Orb_R | Frontal_Sup_Orb_R |
| 10 | Temporal_Inf_R | Angular_R |

## 0.6 Analysis

Previous research has revealed differences in the complex relationships between TCs and subjects with ASD, and that these differences in FC exist in multiple brain regions. To identify which brain regions may have undergone alterations, we leverage the convolutional layer weights in our selected model. This allows us to ascertain which interactions between brain areas have more pronounced impact on classification, thereby suggesting that these areas have important differences between autistic and normal individuals. Examining the network weight of the first layer of the static FC part and the network weight of the second layer of the dynamic FC part, we find that the corresponding weight is a matrix of $32 \times 116$. By taking the absolute value of the weight of the channel and summing it, we can obtain a matrix of $1 \times 116$. we then select the brain areas corresponding to the top 10 features with the largest absolute values as the most discriminative brain areas, as shown in Table 3 and Fig 9 with the help of BrainNet Viewer [47].

As shown in Table 3, we list the top ten regions that contribute most to the classification in static FC and dynamic FC respectively. Among them, the heschl and superior frontal gyrus areas both play an important role in the classification based on two types of FC. We also find that both have a certain proportion in the cerebellum region [48, 49]. Previous classification models or physiological studies also have some similar conclusions that differences exist in these brain areas [50–53]. At the same time, based on the weights coming from the first convolution layer of dynamic FC part, we can also get the smaller frequency bands in which there may be greater differences in the brain activity between subjects with ASD and TCs. After analyzing the weights, we find that the frequency bands between 0.04231Hz-0.04769Hz and 0.06385Hz-0.067436Hz contribute more to classification. Perhaps there are more differences in brain activity between subjects with ASD and TCs in these frequency bands, which requires further physiological experimental confirmation.

Furthermore, in order to elucidate the contributions of SRS scores, static FC and dynamic FC to the model's predictions, we employ the concept of Shapley value [54]. The Shapley value, derived from game theory, is utilized in allocating gains. In the realm of machine learning, Shapley values serve to expound the extent of influence each feature exerts on prediction outcomes within the model. For a given feature set S, the formula for calculating Shapley values is expressed as

$$\phi_S(x) = \sum_{T \subseteq S\ i} \frac{|T|!(|S| - |T| - 1)!}{|S|!} [f(x_T \bigcap i) - f(x_T)], \tag{11}$$

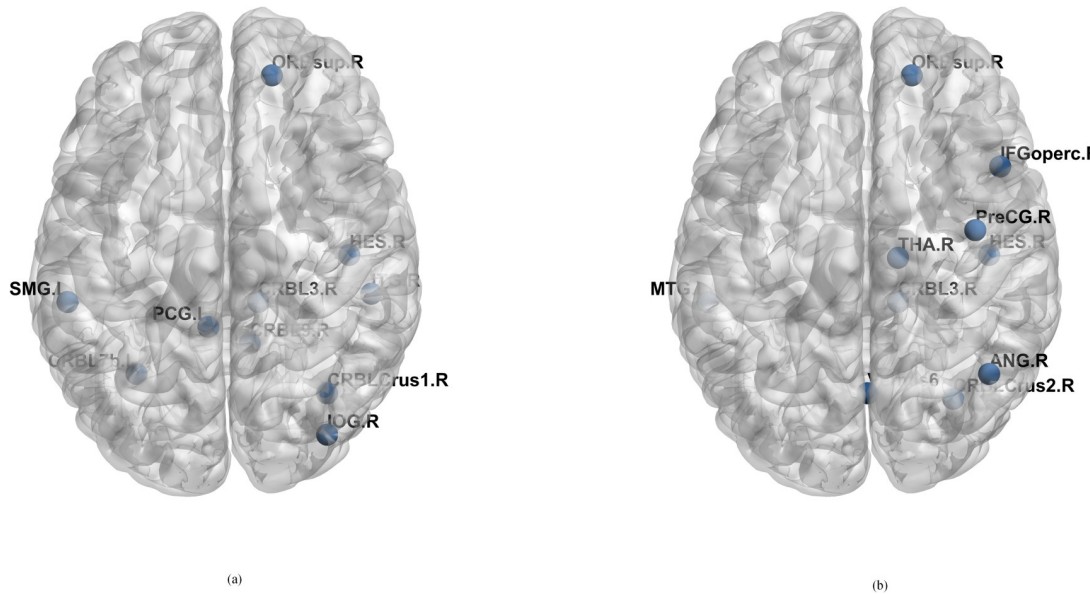

**Fig 9. The most discriminating brain areas related to ASD.**

where i represents the feature to be calculated the Shapley value, T denotes the subset of S except i, $x_T$ signifies the input containing only the features from T, and $x_T \bigcap i$ refers to the input containing features from both T and i. The meaning of this formula is that for any feature i, when it is added to a feature set S, its contribution to the final prediction result is calculated by combining i with other features in S.

In our specific case, for the purpose of comparing the contributions of the three types of inputs, we take the absolute values of the Shapley values computed for each subcategory of features and sum them. We have randomly selected several subjects, and the impact of SRS scores, static FC and dynamic FC on the output is illustrated in the Fig 10. As shown in Fig 10, the prediction of our model is not heavily dependent on one type of inputs.

## Discussion

In pursuit of delineating the physiological disparities between individuals with ASD and TCs, we deploy a synergistic approach that melds both static FC and dynamic FC, defining specialized convolutional kernels for feature extraction from FC matrices. This method facilitates the identification of critical brain regions that are particularly contributory to distinguishing between the two groups–some of which have been corroborated by previous studies. Moreover, we discern two brain regions, the heschl and superior frontal gyrus, that contribute substantially to both static FC and dynamic FC for classifications. Additionally, we pinpoint narrower frequency bands where dynamic FC differences may exist. To further augment the diagnostic capabilities for autism, we incorporate SRS, combining them with FC features to auxiliary diagnose ASD effectively. In this way, our model achieves better classification results compared to some previous papers.

### 0.7 Limitation

Meanwhile, our model also has certain limitations. When applying this model to a new dataset, it can be difficult to obtain social measures. When applying this model to other mental

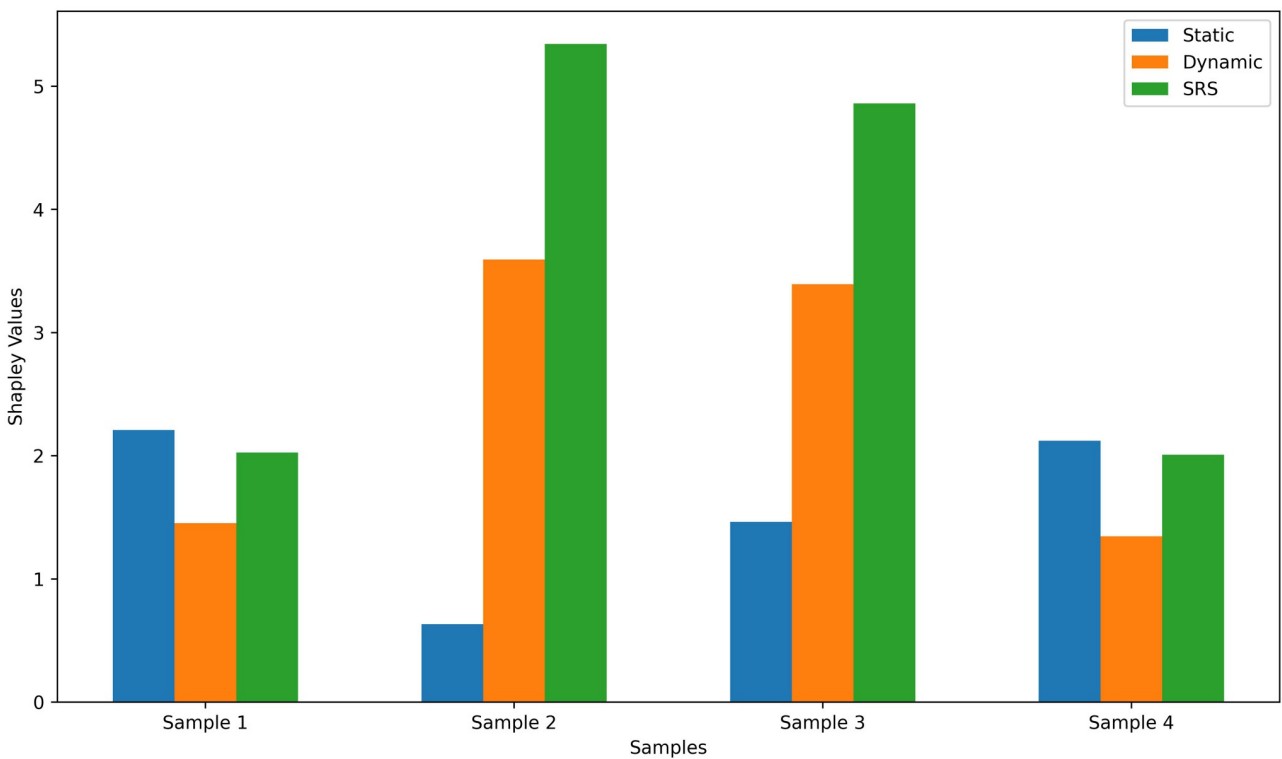

**Fig 10. The Shapley value corresponds to SRS scores, static FC and dynamic FC from random selected subjects.**

illnesses, such as depression, there are different perspectives and understandings when using related measures, which requires corresponding adjustments when applying the model.

Besides, autism has different severity levels, and these different levels of severity may exert an influence on categorization. Regrettably, our model does not reflect this well. Furthermore, our proposed model cannot infer the severity of the participants. In clinical practice, a comprehensive assessment, incorporating tools such as the diagnostic and statistical manual of mental disorders and clinical observations, is employed to definitively ascertain the severity of autism. Regrettably, our model does not have the ability to do this.

## 0.8 Future prospects

For future research, the integration of more diverse data sources for classification could be explored–for instance, by amalgamating various brain templates or utilizing time-frequency analysis methods to obtain additional FC information. Facial information of subjects can also be integrated for autism diagnosis [11]. Another way is merging the text information obtained from communicating with patients and performing natural language processing to assist in screening. The introduction of these information not only advances the prospects of AI in medical diagnostics but also minimizes the likelihood of false positives or identifying those who pretend to be patients. In addition, the proposed model holds potential for aiding the diagnosis of other psychological conditions, such as depression.

Besides, although we use Shapley value as a visual display of model interpretability, there is no similar paradigm or formula that tells us why the model makes such a decision. It does not tell us how the model relates deep features to the inputs. The clinical interpretability of the

model can be figured out. In our research, we find that certain brain regions make a greater contribution to the diagnosis of autism. We also find that some previous articles also find differences in these regions. In clinical practice, further analysis may be possible, such as calculating the FC of these brain areas, and integrating the FC of multiple brain areas to obtain a threshold for distinguishing subjects with ASD from TCs. This requires more experimental subjects to obtain a more generalizable result.

Finally, since ASD is a developmental disorder characterized by potential fluctuations in behavior and brain function over time, longitudinal data could potentially offer additional insights into how ASD evolves over time. Subsequent scholars may consider seeking participants for months or even years of follow-up studies, meticulously documenting changes in their social behaviors and MRI data, thereby facilitating further study.

## Conclusion

In summary, this article offers a valuable insights into autism prediction. The proposed model transcends the confines of a singular data type, presenting a hybrid CNN-SVM network model that combines ASD early screening tools with resting-state brain fMRI data. This model demonstrates good performance in classifying data across different genders, age groups and sites. Moreover, using FC, the model not only identifies brain regions with significant influence on classification outcomes but also elucidates the frequency bands that impact classification. These findings may offer clues to the etiological mechanisms and the determination of biological markers for autism. This research extends the comprehension of autism diagnosis models, encouraging comprehensive predictive analysis through the integration of multifaceted information.

In conclusion, this study's significance lies in the introduction of a hybrid CNN-SVM model that integrates social metrics and fMRI data. It charts a promising method for future research, which is integrating diverse sources of information for prediction.

## Author Contributions

**Conceptualization:** Linjie Qiu.

**Data curation:** Linjie Qiu.

**Formal analysis:** Linjie Qiu.

**Funding acquisition:** Jian Zhai.

**Investigation:** Linjie Qiu.

**Methodology:** Linjie Qiu.

**Project administration:** Jian Zhai.

**Resources:** Linjie Qiu.

**Software:** Linjie Qiu.

**Supervision:** Jian Zhai.

**Validation:** Linjie Qiu, Jian Zhai.

**Visualization:** Linjie Qiu.

**Writing – original draft:** Linjie Qiu.

**Writing – review & editing:** Jian Zhai.

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
