## [Decision Letter · Decision Letter 0]

23 Oct 2023

PONE-D-23-32535A hybrid CNN-SVM model for enhanced autism diagnosisPLOS ONE

Dear Dr. Qiu,

Thank you for submitting your manuscript to PLOS ONE. After careful consideration, we feel that it has merit but does not fully meet PLOS ONE’s publication criteria as it currently stands. Therefore, we invite you to submit a revised version of the manuscript that addresses the points raised during the review process.

Major rework and revision is needed based on the reviewers' feedback and comments appended below.

We look forward to receiving your revised manuscript.

Kind regards,

Umer Asgher, PhD

Academic Editor

PLOS ONE

“This work was supported by the National Natural Science Foundation of China under Grant numbers 11671354.”

“This work was supported by the National Science Foundation of China under Grant numbers 11671354.”

“This work was supported by the National Natural Science Foundation of China under Grant numbers 11671354.”

5. We note that Figures 1 and 6 in your submission contain copyrighted images. All PLOS content is published under the Creative Commons Attribution License (CC BY 4.0), which means that the manuscript, images, and Supporting Information files will be freely available online, and any third party is permitted to access, download, copy, distribute, and use these materials in any way, even commercially, with proper attribution. For more information, see our copyright guidelines: http://journals.plos.org/plosone/s/licenses-and-copyright.

1. You may seek permission from the original copyright holder of Figures 1 and 6 to publish the content specifically under the CC BY 4.0 license.

Reviewers' comments:

Reviewer's Responses to Questions

**Comments to the Author**

1. Is the manuscript technically sound, and do the data support the conclusions?

Reviewer #1: Yes

Reviewer #2: Yes

Reviewer #3: Yes

Reviewer #4: Partly

Reviewer #5: Yes

Reviewer #6: No

2. Has the statistical analysis been performed appropriately and rigorously? 

Reviewer #1: N/A

Reviewer #2: N/A

Reviewer #3: Yes

Reviewer #4: No

Reviewer #5: Yes

Reviewer #6: No

3. Have the authors made all data underlying the findings in their manuscript fully available?

Reviewer #1: Yes

Reviewer #2: Yes

Reviewer #3: Yes

Reviewer #4: Yes

Reviewer #5: Yes

Reviewer #6: Yes

4. Is the manuscript presented in an intelligible fashion and written in standard English?

Reviewer #1: Yes

Reviewer #2: Yes

Reviewer #3: No

Reviewer #4: No

Reviewer #5: Yes

Reviewer #6: No

5. Review Comments to the Author

Reviewer #1: I think the authors need to do some efforts for comparing his approach with recent research studies such as in:

https://onlinelibrary.wiley.com/doi/full/10.4218/etrij.2021-0097

You will find that this paper compares only one AutoML package from all available packages with Transfer Learning and Traditional Machine Learning. In addition, Linear Regression (LR) produce linear models which are similar to SVM and I am wondering why the authors did not compare the performance of the hybrid CNN-LR with CNN-SVM.

Reviewer #2: While the paper presents a novel and promising approach for diagnosing Autism Spectrum Disorder (ASD), it's important to acknowledge the limitations of the work. Some potential limitations of this research could include:

1. Keywords are missing in the abstract.

2. The sample size of 379 subjects with ASD and 442 typical controls, while substantial, may not be representative of the entire ASD population. The diversity of individuals with ASD in terms of age, gender, and severity of symptoms could impact the generalizability of the findings.

3. The accuracy and reliability of the resting-state fMRI data can greatly influence the results. Issues such as motion artifacts, data preprocessing, and the consistency of data collection across different sites could affect the quality of the data.

4. Deep learning models, including CNNs, have a risk of overfitting, especially when working with relatively small datasets. It's crucial to address how the model avoids overfitting and whether any techniques like cross-validation or regularization were used.

5. The model's performance might be optimized for the specific dataset and experimental setup used in the study. The paper should discuss the potential challenges in applying this model to new or different datasets.

6. The paper emphasizes the use of both static and dynamic functional connectivity features alongside SRS metrics. However, the paper should address whether the model's performance is heavily reliant on one type of feature and how generalizable these features are to other ASD populations.

7. While achieving a high classification accuracy is important, it's also crucial to consider the clinical interpretability of the model. Understanding the biological or clinical significance of the identified features and regions is essential for making practical use of the findings.

8. The paper should discuss the interpretability of the CNN-SVM model. Deep learning models are often seen as "black boxes," making it challenging to explain the model's decision-making process.

9. Any potential bias in the data collection, preprocessing, or in the model itself should be addressed. Additionally, the ethical considerations regarding the use of machine learning in healthcare, including issues related to privacy and consent, should be discussed.

10. ASD is a developmental disorder, and it often involves changes in behavior and brain function over time. This study seems to focus on cross-sectional data. Longitudinal data might provide additional insights into how ASD evolves over time.

11. The study should discuss the validation methods employed, such as cross-validation, and the reproducibility of the results to ensure the robustness of the model's performance.

12. The authors used some evaluation metrics but they do not give us a clear understanding of the level of false positives and false negatives directly. Therefore, recently mean Intersection over union (mIoU) has been widely used as a more reasonable and intuitive way of metrics. mIoU should be highlighted in the Abstract and conclusion.

13. The proposed work must be compared with some state-of-the-art works from 2022/2023 for better understanding the capabilities of the proposed work.

Reviewer #3: 1. Typing and grammatical errors are highlighted in the attached file

2. Unify all over the text Figure or Fig.

• 70 Figure 1.

• 175 Fig 2.

• 237 Fig 4 illustrates the performance of three different

3. Figure caption is not a sentence:

• Fig. 5 This figure compares…..

A possible caption can be:

"Performance of the proposed model compared with related work

(Wang [22]; Huang [23]; Yin [24]; Jiang [25]; and Bhandage [26])"

• Fig 6. This figure shows the most discriminating brain areas.

A possible caption can be:

"The most discriminating brain areas related to ASD"

• Punctuation ruled should be respected.

Reviewer #4: Author presented a hybrid approach for autism spectrum disorder diagnosis. They made use of CNN and SVM based model to achieve this. But, there are several drawbacks in the article that should be resolved.

1. Abstract is not clear. It should express research backround , problem statement, methods and results.

2. Introduction section failed to mention the challenges of existing system, major contribution and structure of the manuscript.

3. Literature review is poor. Several works should be studied such as Deep Learning for Autism Diagnosis and Facial Analysis in Children, Conditional Generative Adversarial Network Approach for Autism Prediction, Deep learning-based feature selection and prediction system for autism spectrum disorder using a hybrid meta-heuristics approach, Machine learning for autism spectrum disorder diagnosis using structural magnetic resonance imaging: Promising but challenging.

4. Proposed model can be expressed as an algorithm. Architecture of hybrid model can be represented.

5. What is the need for devising hybrid model?. Mention the significance of incorporating SVM with CNN model.

6. Several dataset can be used for experimentation. Hyper parameter tuning needs to be explained.

7. Discussion section can be included to state the challenges, inferences and future prospects.

8. Conclusion section can be improved.

Reviewer #5: The contributions is clear. However, authors should justify the decisions that made in the article such as the selected methods and parameters settings. The authors should state the parctical implications of the work.

Reviewer #6: The following points need to be addressed in the revision:

1. The abstract is adequate in length and structure.

2. The indication of the dataset in the abstract must be ensured.

3. Comparison with SOTA techniques should be mentioned in the abstract

4. The literature review of the study is insufficient, there are many important studies that are missing. The authors should include the following references with proper discussions:

https://doi.org/10.1109/iCareTech49914.2020.00032

https://doi.org/10.3390/app12083715

https://doi.org/10.3390/medicina58081090

https://doi.org/10.1038/s41598-023-30309-4

https://doi.org/10.1088/1742-6596/1916/1/012226

https://doi.org/10.1088/1757-899X/1055/1/012115

https://doi.org/10.1109/INDISCON50162.2020.00037

https://doi.org/10.1016/j.fcij.2017.12.001

https://doi.org/10.1016/j.cmpb.2019.05.015

5. Please add contributions of your work in bulleted form preferably in the introduction section

6. The organization of the article is missing.

7. There is irregular text in the article Lines: 62-63. Ensure a consistent heading number scheme. Correct it.

8. How you are claiming CNN static, rather the converse is true.

9. Defining static FC, you cannot change the working principle of CNN. Changing the name of the remarkable technique is no good. Please justify your approach.

10. Under the heading of Neural Network, do not use CNN. One is shallow and the other is deep.

11. Table 1 is without any whereabouts. The column headings, if possible, may be provided.

12. The distribution of instances in the dataset should be graphically represented.

13. The figures related to CNN are not clearly represented according to the norms of deep learning. No figure number is there. Figures are contradictory and do not truly represent the working principle of algorithms.

14. Bhandage has almost the same results as yours. Justify that the proposed solution would always be guaranteed through some statistical analysis.

15. Limitations of the proposed work with future recommendations should be added before the conclusion.

16. The Conclusion needs rephrasing, as it is not encircling the proposed work being asked for publication.

6. PLOS authors have the option to publish the peer review history of their article (what does this mean?). If published, this will include your full peer review and any attached files.

Reviewer #1: No

Reviewer #2: No

Reviewer #3: **Yes: **Said Ghoniemy; AinShams University

Reviewer #4: **Yes: **No

Reviewer #5: No

Reviewer #6: No

---

## [Author Response · Author response to Decision Letter 0]

4 Jan 2024

Dear Editor,

 Thank you for your giving us an opportunity to revise our manuscript! Now we are submitting the revised manuscript entitled "A hybrid CNN-SVM model for enhanced autism diagnosis" for consideration for publication in PLOS ONE.

 In the revision, we have performed nearly all the suggested opinions, experiments and analyses, and fully addressed the comments made by the reviewers and editor. The finished new experiments and data analyses as following: 

 1. We have revised the Abstract and Results sections to meet the requirements required for publication. And we add more related works in the Introduction section to review the application of machine learning in medical aspects, especially in the diagnosis of autism.

 2. We have added the contributions of this work and the organization of this manuscript. We also have described the flowchart in algorithmic form to enhance the comprehension of our model.

 3. In order to better evaluate the proposed model, we have introduced more indicators to evaluate the performance of different classifiers in autism diagnosis, as shown in the Fig 5.

 4. We have analyzed the performance of the proposed model on different genders, sites, and age groups, and the results are shown in Figs 6-8.

 5. We also used the Shapley value to measure the impact of three types of inputs on model predictions, giving interpretability to a certain extent.

 6. We have added Limitation and Future prospects to the discussion section for better summary.

 All authors have read and approved the re-submission of the manuscript! If you have any questions, please let me know!

 Thank you for your consideration of our paper and we are looking forward to hearing from you!

 Sincerely yours,

 Linjie, Ph.D., 

 School of Mathematical Sciences, Zhejiang University, Hangzhou, Zhejiang, China

 Email:11935034@zju.edu.cn

---

## [Decision Letter · Decision Letter 1]

12 Feb 2024

PONE-D-23-32535R1A hybrid CNN-SVM model for enhanced autism diagnosisPLOS ONE

Dear Dr. Qiu,

Thank you for submitting your manuscript to PLOS ONE. After careful consideration, we feel that it has merit but does not fully meet PLOS ONE’s publication criteria as it currently stands. Therefore, we invite you to submit a revised version of the manuscript that addresses the points raised during the review process.

Specifically; Two reviewers have accepted the manuscript after re-evaluation and One reviewer has give minor reviews (Feedback),. So please improve the manuscript and re-submit as soon as possible in the light of reviewers comments. 

We look forward to receiving your revised manuscript.

Kind regards,

Umer Asgher, PhD

Academic Editor

PLOS ONE

Journal Requirements:

Reviewers' comments:

Reviewer's Responses to Questions

**Comments to the Author**

1. If the authors have adequately addressed your comments raised in a previous round of review and you feel that this manuscript is now acceptable for publication, you may indicate that here to bypass the “Comments to the Author” section, enter your conflict of interest statement in the “Confidential to Editor” section, and submit your "Accept" recommendation.

Reviewer #3: All comments have been addressed

Reviewer #4: All comments have been addressed

Reviewer #6: All comments have been addressed

2. Is the manuscript technically sound, and do the data support the conclusions?

Reviewer #3: Yes

Reviewer #4: Yes

Reviewer #6: Yes

3. Has the statistical analysis been performed appropriately and rigorously? 

Reviewer #3: Yes

Reviewer #4: Yes

Reviewer #6: No

4. Have the authors made all data underlying the findings in their manuscript fully available?

Reviewer #3: Yes

Reviewer #4: Yes

Reviewer #6: Yes

5. Is the manuscript presented in an intelligible fashion and written in standard English?

Reviewer #3: Yes

Reviewer #4: Yes

Reviewer #6: Yes

6. Review Comments to the Author

Reviewer #3: All comments were revised and the answers are acceptable. The paper is much enhanced ans is suitable for publication

Reviewer #4: After evaluating the article, I have observed significant modifications. All the comments were addressed. Hence, this article can be considered for publication.

Reviewer #6: All the points have been thoroughly addressed.

The following points are suggested as minor revisions:

1. In line number 80 (page 37), replace “…SRS scores, which reflect the…” with “….SRS scores reflecting the…”.

2. The features that we get from CNN are called dynamic features. In your work, this seems ambiguous, and one way to remove it is to change “static feature” in figure 1 (and throughout the article) to “staticFC features”, and “dynamic feature” to “dynamicFC features”

3. Remove the text lines in the conclusions “Nevertheless, the article harbors limitations, such as its inability…… diagnosis of autism.” You have already mentioned this text in the limitations and future recommendations.

7. PLOS authors have the option to publish the peer review history of their article (what does this mean?). If published, this will include your full peer review and any attached files.

Reviewer #3: **Yes: **Said Ghoniemy, Ain Shan\\ms University, Cairo, Egypt

Reviewer #4: **Yes: **KANNIMUTHU SUBRAMANIAN

Reviewer #6: No

---

## [Author Response · Author response to Decision Letter 1]

28 Feb 2024

Dear Editor,

Thank you for your giving us an opportunity to revise our manuscript! Now we are submitting the revised manuscript entitled ”A hybrid CNN-SVM model for enhanced autism diagnosis” for consideration for publication in PLOS ONE.

In the revision, we have performed nearly all the suggested opinions and fully addressed the comments made by the reviewers and editor. We summarized the modifications as follows:

1. We have checked all references and can ensure that none of them have been withdrawn.

2. We have adjusted the images using the Preflight Analysis and Conversion Engine (PACE) digital diagnostic tool to ensure they meet PLOS requirements.

3. We have made appropriate revisions to the content based on the reviewers’ comments.

All authors have read and approved the re-submission of the manuscript! If you have any

questions, please let me know!

Thank you for your consideration of our paper and we are looking forward to hearing from

you!

Sincerely yours,

Linjie, Ph.D.,

School of Mathematical Sciences, Zhejiang University, Hangzhou, Zhejiang, China

Email:11935034@zju.edu.cn

---

## [Editor Report · Decision Letter 2]

1 Apr 2024

A hybrid CNN-SVM model for enhanced autism diagnosis

PONE-D-23-32535R2

Dear Dr. Qiu,

We’re pleased to inform you that your manuscript has been judged scientifically suitable for publication and will be formally accepted for publication once it meets all outstanding technical requirements. There are minor issues to be addressed in the light of reviewers comments.

Kind regards,

Umer Asgher, PhD

Academic Editor

PLOS ONE